# Enhancing Cellular Uptake of Native Proteins through Bio-Orthogonal Conjugation with Chemically Synthesized Cell-Penetrating Peptides

**DOI:** 10.3390/pharmaceutics16050617

**Published:** 2024-05-03

**Authors:** Jekaterina Nebogatova, Ly Porosk, Heleri Heike Härk, Kaido Kurrikoff

**Affiliations:** Institute of Technology, University of Tartu, Nooruse 1, 50411 Tartu, Estonia

**Keywords:** protein therapeutics, protein transduction, cell-penetrating peptides, bio-orthogonal conjugation

## Abstract

The potential for native proteins to serve as a platform for biocompatible, targeted, and personalized therapeutics in the context of genetic and metabolic disorders is vast. Nevertheless, their clinical application encounters challenges, particularly in overcoming biological barriers and addressing the complexities involved in engineering transmembrane permeability. This study is dedicated to the development of a multifunctional nanoentity in which a model therapeutic protein is covalently linked to a cell-penetrating peptide, NickFect 55, with the objective of enhancing its intracellular delivery. Successful binding of the nanoentity fragments was achieved through the utilization of an intein-mediated protein-trans splicing reaction. Our research demonstrates that the fully assembled nanoentity-containing protein was effectively internalized by the cells, underscoring the potential of this approach in overcoming barriers associated with protein-based therapeutics for the treatment of genetic disorders.

## 1. Introduction

Protein transduction, especially the intracellular delivery of therapeutic proteins, has the potential to improve the treatment of genetic and metabolic disorders [1,2,3,4,5,6]. By delivering therapeutic proteins directly to specific cell types or organelles, this approach enables highly precise treatment. Compared to small-molecule drugs, native proteins offer numerous advantages, including significantly higher bioactivity, lower toxicity, and greater resistance to degradation in the bloodstream [7]. However, in order to retain these benefits and remain biologically active, the proteins must be correctly folded and undergo necessary modifications [8]. 

While the majority of the current research focuses on developing scalable systems for industrial production, the bacterial systems traditionally used for this purpose are often unable to replicate eukaryotic molecules entirely. In contrast, mammalian cells, particularly those derived from human tissue, offer an environment that closely resembles human physiology, rendering them an ideal choice for producing and testing proteins intended for use in humans. The conditions within the mammalian cells, as well as the post-translational modifications that occur after protein synthesis, significantly impact protein structure, function, and regulation. These modifications play a crucial role in various aspects of protein activity, including signaling, enzymatic activity, stability, and protein localization. Essential modifications include glycosylation for bioactivity, stability, and immunogenicity [9]; phosphorylation for signaling and regulation [10]; disulfide bond formation for maintaining structural integrity and functionality [11]; proteolytic cleavage for correct protein activation [12]; and proper folding to ensure functionality and avoid aggregation [13]. 

Despite the potential benefits of therapeutic proteins, their efficient implementation may face certain challenges related to their physical and chemical properties. These molecules may encounter obstacles due to their size and charge, which limit their ability to overcome biological barriers. Additionally, once the proteins are delivered, they may be vulnerable to degradation or loss of activity within the cell, ultimately impacting their therapeutic efficacy [14]. Endocytic delivery pathways, for example, may result in endosomal entrapment and potential degradation, highlighting the need for endoplasmic agents to facilitate efficient cargo release [15,16,17]. 

The efficiency of transduction relies heavily on the specific cell type and cargo, which makes achieving consistent and high efficiency challenging in a variety of contexts [18]. Although transduction can be successful in the cell culture, in vivo delivery of transduced proteins is often hindered by obstacles such as tissue penetration, systemic stability, and clearance, which must be overcome for successful clinical applications [19,20,21,22]. One possible solution to these challenges is the implementation of cell-penetrating peptides (CPP) in drug delivery systems [23,24,25,26]. 

This study aims to create a nanoentity that can enhance the cell-penetrating abilities of cargo proteins. To achieve this, we have developed a bio-orthogonal conjugation method that involves combining chemically synthesized CPP with proteins produced in mammalian cell culture. The method presented in this work is based on NickFect 55 (NF55), which is an analog of a well-known amphipatic cell-penetrating peptide transportan 10. The structure of CPP contains a fatty acid modification, a stearoyl tail at the N-terminus, and a branched structure at the side chain of non-proteogenic amino acid ornithine [27,28]. Such modifications improve the key properties of CPP, providing, as a result, lower toxicity and efficient transfection or transduction [29]. Compared to PepFect peptides, the NickFect family of CPPs are also able to address the issue of endosomal escape, which is crucial in achieving biological activity of cargo inside the cell [30].

CPPs are widely recognized as effective mediators for delivering a variety of cargo molecules, such as nucleic acid, proteins, lipids, and peptides, across cell membranes, both in vitro and in vivo [27]. When it comes to nucleic acids, complexation through non-covalent association is not an issue since their bioactivity is often encoded in their primary sequence. However, for proteins with diverse pool of structures, covalent attachment becomes necessary, and this process must occur in a biological environment to ensure that protein’s native folding is preserved. This aspect limits the number of bioconjugation methods that can be utilized [24].

The process of bioconjugation is essential in linking proteins produced in mammalian cells with chemically synthesized CPPs. Each method has strengths and limitations, which are outlined in Table 1. For our project, we have chosen to utilize a protein-trans splicing (PTS) technique for bioconjugation. PTS is highly versatile and has numerous applications in protein engineering and chemical biology due to the unique pathway for the modification and cleavage of precursor proteins [31]. Protein splicing involves the modification of a protein precursor by removing the intein (intervening protein) and linking the exteins (external proteins) via a peptide bond to form the mature protein structure. Here, we have used a strategy of splitting the intein sequence into two fragments, intein^N^ and intein^C^, which are, respectively, located at the N- and C-termini. We designed the ligated exteins in such a way that they originate from the two separate precursor proteins. In our approach, one of these counterparts is chemically synthesized and linked with the CPP, while the other is produced biologically and fused to a model protein. Our main aim—and a significant benefit of this technique—is that the reaction is entirely biocompatible, meaning it can occur in a biological system without impacting the protein’s higher-order structure or bioactivity. 

## 2. Materials and Methods

### 2.1. Cell Culture Maintenance

Adherent CHO-K1 (ECCC, CHO 85050302), Neuro2a (ATCC, CCL-131), HeLa (ATCC, CCL-2), and HEK293 (obtained from Prof. Andres Merits) cells were grown in Dulbecco’s Modified Eagle Medium (DMEM), supplemented with 0.1 mM non-essential amino acids, 1.0 mM sodium pyruvate, 100 U/mL penicillin, and 100 mg/mL streptomycin. A total of 10% (final) fetal bovine serum (FBS) was added to complete the media. Cells were passaged regularly when the confluence of cells reached 80–90%. Cells were maintained in a humidified incubator with 5.0% CO_2_ at 37 °C.

HEK293FT cells (obtained from Dr. Alla Piirsoo, Thermo Fisher Scientific; catalog number: R70007) were grown as a suspension cell culture in HEK TF media (Sartorius Xell GmbH, Bielefeld, Germany), supplemented with 100 u/mL penicillin, 100 mg/mL streptomycin, and 6 mM Glutamax. Cell viability was assessed daily, and cell density was regularly reduced. Cells were maintained in a humidified incubator with 8.0% CO_2_ at 37 °C. 

### 2.2. Reporter Plasmid Design and Cloning

Split intein sequences were obtained from the previously published work of David et al. (2015) [41]. Plasmids that contained the inserts of the split intein sequence were ordered from General Biosystems (General Biosystems, Inc., Durham, NC, USA). Cloning was performed using Thermo Fischer (Thermo Fischer Scientific, Waltham, MA, USA) and Solis BioDyne (Solis Biodyne, Tartu, Estonia) reagents. Plasmids used in this work are listed in Appendix A.

### 2.3. Peptide Synthesis

Cell-penetrating peptide NF55 with extended C-terminus was synthesized on an automated peptide synthesizer (Biotage Initiator+ Alstra, Uppsala, Sweden) using the fluorenylmethoxycarbonyl protecting group (Fmoc)-based solid-phase peptide synthesis strategy with Rink-amide ChemMatrix resin to obtain the C-terminally amidated peptide. DIC and Oxyma were used as coupling reagents and DIEA as activator base. The fatty acid (5 eq) and carboxytetramethylrhodamine (1 eq) were coupled manually to the N-terminus of the peptide at room temperature overnight using the aforementioned coupling reagents. The synthesized peptide was cleaved from the resin and purified with preparative HPLC (column Agilent Zorbax 300SB-C3, 5 μm, 250 × 9.4 mm) using a gradient of H_2_O and acetonitrile containing 0.1% of TFA. The molecular weight of the peptide was analyzed via matrix-assisted laser desorption–ionization and time-of-flight mass spectrometry (Bruker Daltonics GmbH and Co. KG, Bremen, Germany), and the purity of the peptide was determined with UPLC (column ACQUITY UPLC BEH130 C18, 1.7 μm, 100 × 2.1 mm) using a gradient of H_2_O and acetonitrile containing 0.1% of TFA. Intein^C^ peptide with azide group was synthesized by Biosynth (Biosynth Ltd., Compton, UK).

### 2.4. Cu-Catalyzed Azide–Alkyne Cycloaddition for Intein-Activated CPP NF55 Synthesis

For the conjugation of modified NF55 and intein^C^, Cu-catalyzed azide–alkyne cycloaddition was used according to the modified protocol previously published by Presolski et al. [42]. Briefly, 25 μL of 4 mM NF55XK(Rhod)XPra in water was diluted in 407.5 μL of HEPES buffer (pH 7.0) and mixed with 10 uL of 20 mM intein^C^–N_3_, where X—aminohexanoic acid (used as a linker); Pra—propargylglycine; Rhod—Carboxytetramethylrhodamine; and N_3_—azide group. In a separate tube, 20 mM CuSO_4_ in water was mixed at 1:2 with 50 mM tris (benzyltriazolylmethyl) amine (TBTA) in DMSO. Next, 7.5 μL of CuSO_4_/TBTA mixture, 25 μL of 100 mM aminoguanidine hydrochloride in water, and 25 μL of 100 mM sodium ascorbate in water were added to the reaction mixture containing NF55XK(Rhod)Xpra and intein^C^–N_3_ and mixed thoroughly. The reaction was allowed to proceed for 24 h at room temperature with continuous shaking. After the click reaction, the presence of NF55–XK(Rhod)X–intein^C^ conjugate was determined with UPLC, the mixture was purified with preparative HPLC, and the conjugate was lyophilized to be used for further experiments.

### 2.5. Transfection and Fusion Protein Production

For protein production in mammalian cells, we utilized a protocol previously optimized for our laboratory setup [43]. Briefly, HEK293FT or CHO-K1 suspension cells were transfected with pDNA-encoding fusion proteins with Hisx6 tag. Transient transfection of mammalian cells was performed via co-incubation with pre-formed CPP/pDNA complexes. Complex components were mixed in ultrapure water, where 2 μL of 1 mM NF55 peptide per 1 μg of pDNA was added. The complex was formed in 10% of the final transfection volume (e.g., 25 μL per 250 μL of transfection mixture). Mixture containing transfection complexes was incubated for 20–40 min at room temperature before addition to the cells to allow for complex formulation. Next, complexes were added to the cell culture. Cells were harvested at a 48-to-72-h time point. The level of fluorescence of reporter protein was measured with a fluorimeter (SynergyMx, BioTek, Winooski, VT, USA), and flow cytometry was performed using an Attune™ NxT Flow Cytometer (Thermo Fischer Scientific, Waltham, MA, USA) equipped with a 488 nm argon laser. The viable cell population was determined from a plot of forward-scattered (FSC) vs. side-scattered (SSC) light. Attune™ NxT Software 3.2.1 software was used to analyze a minimum of 10, 000 events per sample. The area scaling factor was 1.28, with a threshold of SSC 0.3, FSC 220, SSC 360, BL1/GFP 460, and RL1/APC/Cy5 520.

### 2.6. Flow Cytometry 

A total of 40,000 HeLa or Neuro2a cells were seeded on a 24-well plate in 500 μL of medium. The next day, cells were coincubated with the nanoentity for 4 h and then analyzed via flow cytometry. The percentages from the live cell population are shown on each graph. EGFP+ shows gated events with signal intensities over the threshold of ~1% of untreated cells. Flow cytometry was performed using an Attune™ NxT Flow Cytometer equipped with a 488 nm argon laser. The viable cell population was determined as stated above.

### 2.7. Confocal Microscopy

For confocal microscopy, 40,000 adherent HeLa or Neuro2a cells were seeded on an 8-well Nunc™ Lab-Tek™ (Thermo Fischer Scientific, Waltham, MA, USA) and incubated for 24–48 h. Next, cells were either transfected with 0.1 µg of the plasmid or transducted with the nanoentity. Cells were washed, and media were replaced with phenol red-free DMEM prior to visualization. Confocal images were captured from live cells with Zeiss LSM710 (Carl Zeiss AG, Germany) or Zeiss LSM900 (Carl Zeiss AG, Oberkochen, Germany). The images were taken with 10×, 20×, or 63× magnification. For analysis, Zen 2012 software was used.

## 3. Results

### 3.1. Design of Nanoentity for Intracellular Protein Delivery

In order to optimize the intracellular protein cargo delivery, we designed a nanoentity in which a model protein was covalently attached to a cell-penetrating peptide (CPP) through a protein trans-splicing (PTS) reaction (Figure 1). It should be noted that all expression plasmids utilized in this study were specifically designed for employment within mammalian cell systems.

For our proof-of-concept study, we utilized EGFP, a naturally occurring fluorescent protein, as a model protein component of the nanoentity. We selected this protein for its qualities and rapid maturation in the cell. To facilitate the intracellular delivery of the nanoentity, we utilized CPP NickFect 55 (NF55) as an effective mediator for the transmembrane delivery of various biological macromolecules [28,46] (Table 2).

In order to link the substrate fragments of the nanoentity, we utilized the bioconjugation method PTS, where product from the precursors was linked via a peptide bond. To accomplish this, we utilized an intein sequence that could be split into fragments and is known for quick maturation and reaction. The PTS reaction is initiated by two independent and separate fragments: intein^C^ and intein^N^ (Table 2). The intein^N^ component is significantly longer and would pose a significant challenge if one wished to chemically synthesize it. However, we took advantage of the *Nostoc* sp. intein^C^ as this has the additional advantage of having a notably short primary sequence; it is only 39 amino acids long, and it is possible to achieve chemical synthesis via traditional automated solid-phase coupling. We chemically synthesized NF55 and conjugated it with intein^C^ via click chemistry, producing intein-activated NF55. The biologically produced EGFP protein is fused with intein^N^ via molecular cloning and was bioproduced in mammalian cells.

After successful generation of the fragments, the subsequent step involves bio-orthogonal conjugation to attach the CPP to the protein. Our design specifies that this conjugation is facilitated by intein trans-splicing reaction. In the presence of both compounds in the mixture, the intein^C^ and intein^N^ interact and form a HINT (hedgehog/intein-like) structure. This initiates the next reaction and the intein undergoes self-excision, ultimately resulting in the substrate fragments being covalently linked via a peptide bond (Figure 1).

### 3.2. Split EGFP Assay for Bioconjugation Method Validation in Mammalian Cells 

We evaluated the efficiency of the PTS bioconjugation method in mammalian cells using a split EGFP assay. We designed two sets of plasmids (Table 2) that included fusion protein sequences with EGFP protein, splitting the full protein primary sequence between them. Consequently, the expressed fusion proteins from different plasmids do not emit a reporter green fluorescence signal individually unless combined into full protein. The EGFP acquires its fluorescent properties only when both fragments of the protein are expressed and properly joined together, forming a mature functional protein. 

To verify the successful intein trans-splicing with our set of chosen inteins, and to eliminate doubt that EGFP spontaneously self-assembles its subunits, we conducted the split EGFP in a more challenging setup whereby the EGFP was split randomly, outside its subunits. In this way, fully functional fluorescent protein can be re-assembled only via covalent conjugation of the fragments, which can only occur via an intein reaction. As a first approach, to validate the conjugation in the biological environment, each plasmid contained a split intein fragment, intein^N^ or intein^C^, respectively, and an additional fluorescent reporter protein to monitor each plasmid individually. Reporter proteins mCherry (red) or SBFP (blue) were utilized for this purpose as individual expression control reporters.

When the split intein fragments from both plasmids are expressed in a single cell, they form a mature intein structure and undergo intein self-excision, resulting in the ligation of the split EGFP fragments via peptide bond formation (Figure 2a, Table 3).

The split EGFP fragments (Table 3) were introduced into mammalian cells via transfection of the plasmids, which allows for the expression of the individual plasmid products in these cells. When either plasmid was expressed alone, only mCherry (red) or SBFP (blue) reporter protein was produced; the green signal was absent as the full EGFP could not be formed as neither of the fragments were present (Figure 2b). However, when both plasmids were simultaneously used in a double-transfection manner, both EGFP fragments were expressed in the same cell, resulting in the detection of both expression controls mCherry and SBFP reporter proteins, as well as the green signal from the assembled full EGFP protein (Figure 2 and Appendix A), indicating successful intein trans-splicing reaction. We confirmed the expression of both EGFP fragments and the successful reaction of intein trans-splicing by observing the green reporter signal of the mature EGFP protein in the cells (Figure 2b). 

Another set of plasmids was designed and validated to confirm assay validity. The two sets of plasmids differ in the way the EGFP sequence is divided: the first set includes plasmids where one encodes the first ten subunits of EGFP and the second encodes the eleventh subunit of the protein; the second set of plasmids has randomly divided EGFP sequence fragments (Table 3, Appendix A). 

We verified the successful PTS reaction by monitoring the combined expression from both fragments simultaneously, as well as the formation of green reporter. Together, the above findings confirm that the formation of a full and functional mammalian protein was successfully achieved in the biological environment via a PTS reaction. 

### 3.3. Synthesis of the Nanoentity Substrate Compounds

#### 3.3.1. Substrate 1: Biosynthesis of EGFP–Intein^N^ Fusion Protein

Next, we sought to produce a model protein that could be delivered into the cells. We chose EGFP, which is a good model protein because it is widely used, easy to detect, stable, and versatile. EGFP–Intein^N^ fusion protein, additionally containing Hisx6-tag for the purification purposes, was designed, produced in mammalian cells, and purified (Figure 3a,b). We assessed and optimized the production of our protein in HeLa, CHO-K1, and HEK293FT cells. Recently, we have shown that suspension cell culture introduction can significantly improve the yield of protein production [43]. Therefore, we chose suspension cells for production and have successfully produced and purified EGFP–Intein^N^.

#### 3.3.2. Substrate 2: Chemical Synthesis of Intein^c^-activated NF55 Peptide

We utilized the CPP NF55 as the second component of the nanoentity and the actuator of the cellular entry (Figure 3c). Notably, NF55 is a highly efficient transcellular delivery agent, but as it contains non-canonical amino acids and a stearoyl tail, it cannot be biologically produced, leaving chemical synthesis as the only option. Additionally, the NF55 moiety includes the intein^C^ sequence and a fluorescent label for tagging purposes. Therefore, the synthesis of the chemically synthesized nanoentity substrate was achieved in several steps:(1)NF55, with extended C-terminus to include propargylglycine for later click reaction, was synthesized and labeled with carboxytetramethylrhodamine (Rhod).(2)Intein^C^ peptide was designed to contain azide group.(3)The fluorolabeled NF55 and azide–intein^C^ peptide were then conjugated through copper-catalyzed azide–alkyne click reaction. The successful formation of the nanoentity substrate was confirmed via UPLC (Appendix A).

To confirm the cell-penetrating properties of the chemically synthesized nanoentity substrate, we incubated NF55–Rhod–intein with mammalian cells and assessed its intracellular distribution through the attached fluorescent label. Confocal microscopy images revealed that the chemical compound of the nanoentity can enter the cells without cargo (Figure 3d).

### 3.4. Assembly of the full Nanoentity and Protein Transduction in Mammalian Cells

Upon successful synthesis of both nanoentity substrates, we proceeded to perform the bioconjugation of EGFP–Intein^N^ and intein^C^-activated NF55 in vitro (Figure 1). The reaction protocol was developed by optimizing various factors, such as buffer composition, concentration of the substrate compounds, reaction temperature, and duration. After identifying the optimal conditions, the reaction was carried out, and the successful conjugation of the full nanoentity was confirmed by changes in peaks detected with UPLC (Appendix A).

Subsequently, we evaluated whether the functional protein of interest was transduced into the cell after confirming the nanoentity conjugation. To do so, we coincubated the nanoentity with mammalian cell lines HeLa and Neuro2a. The cellular uptake efficacy was assessed through confocal microscopy and flow cytometry analysis (Figure 4a,b, Appendix A). With both of these methods, an EGFP signal was detected in the cells, indicating successful transduction of a full, membrane-impermeable protein into mammalian cells.

## 4. Discussion

Our primary objective in this study was to develop a multifunctional nanoentity that can efficiently deliver proteins intracellularly. To accomplish this, we employed a bioconjugation method called protein trans-splicing (PTS), which involves the covalent binding of a protein to a cell-penetrating peptide (CPP) using a specific split intein sequence. This approach could have numerous applications in biotechnology and medicine.

We utilized a reporter protein EGFP as a model protein, and a cell-penetrating peptide, NickFect 55 (NF55), as the trans-membrane transporter. These components were combined using a rapid split intein method. By linking two precursor components through PTS, we were able to achieve a high yield of the desired product. The sequence of the split intein fragments was adapted from previous research conducted by Shah et al. [47], with the intein^N^ and intein^C^ fragments being derived from the DNA polymerase III subunits alpha of different organisms. This particular pair of split intein fragments had previously been shown to facilitate rapid PTS reactions, making it an ideal choice for our design.

Strategically selected split intein fragments were utilized to connect precursor fragments of the nanoentity, addressing the challenge of synthesizing lengthy peptide sequences. NF55 is chemically synthesized as the stearoyl group, non-canonical ornithine amino acid, and the unconventional placement of the peptide bond in the peptide backbone excludes the possibility of biosynthesizing this moiety. Shorter (39-amino-acids) intein^C^ was linked to chemically synthesized NF55, and longer (80- amino-acids) intein^N^ was fused with biologically produced EGFP.

Using a split EGFP assay, we conducted a proof-of-concept study to verify the efficacy of the PTS bioconjugation method in mammalian cells. Our findings showed that EGFP signal emission was only possible if both fragments were expressed together in a single cell, confirming the successful protein trans-splicing process. These results were further supported by confocal microscopy images, which revealed the absence of an EGFP signal in the cells that were transfected with only one construct, and its presence in double-transfected cells, a finding that was corroborated by control reporter signals.

Next, EGFP–Intein^N^ fusion protein was designed, produced, and purified. After optimizing production across various cell types, HEK293FT cells were found to be the most efficient for this specific protein expression.

Additionally, the synthesis of intein^C^-activated NF55 peptide faced challenges due to NF55’s non-canonical amino acid, leading to a stepwise synthesis approach. The confirmed cell-penetrating properties of the chemical compound of the nanoentity were demonstrated through confocal microscopy.

A well-crafted reaction protocol was utilized to achieve successful in vitro conjugation of the nanoentity components. This protocol underwent optimization for buffer composition, compound concentration, temperature, and duration. The efficacy of transduction in CHO-K1 and Neuro2a cells was evident in confocal microscopy images and flow cytometry analysis, showcasing the entry of the bioconjugated nanoentity into cells.

Our research represents a significant advancement in the field of intracellular cargo delivery, offering a robust and versatile nanoentity. The successful assembly, validation, and synthesis of components open avenues for future research. Future studies may include examining the nonentity’s performance with various cargo proteins, optimization of the transduction efficacy, and exploring in vivo applications for a comprehensive understanding of its potential.

In conclusion, our study presents the development of a nanoentity for intracellular protein delivery. By combining innovative bioconjugation methods with established molecular tools, this nanoentity holds promise for transformative applications in various biological and medical contexts. Acknowledging the study’s limitations, such as the need for further nanoentity characterization and in vivo validation, we propose these findings as a stepping stone for future advancements of targeted intracellular cargo delivery.

## Figures and Tables

**Figure 1 pharmaceutics-16-00617-f001:**
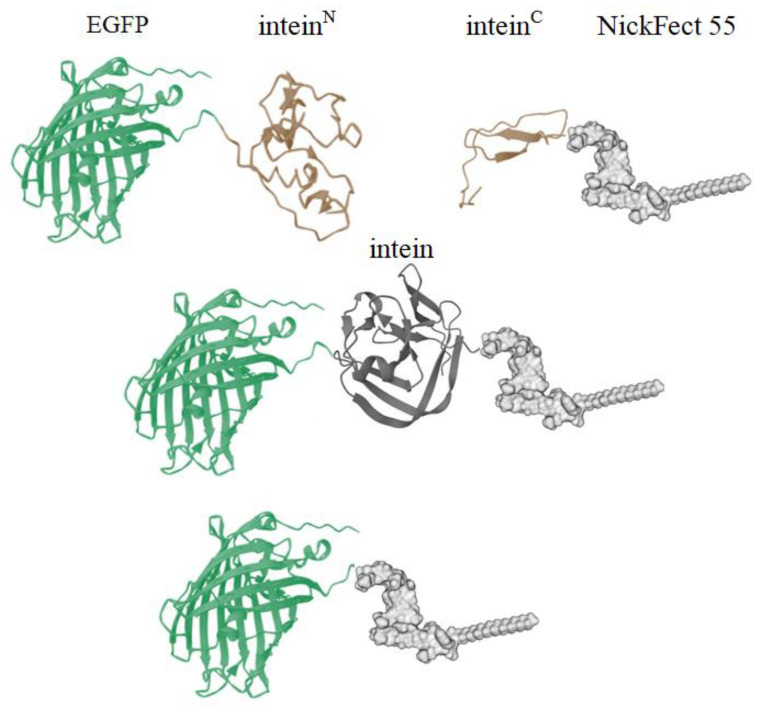
Conjugation schema of the fusion of the intein-activated protein and the intein-activated CPP. The protein is fused to the intein^N^ sequence. When the intein-containing substrates are coincubated in a biological environment, the mature intein HINT structure is formed, which initiates the protein trans-splicing reaction. Once the intein undergoes self-excision, the protein and CPP are joined together through a peptide bond. Structures were generated using Mol* [44] software and Marvin [45].

**Figure 2 pharmaceutics-16-00617-f002:**
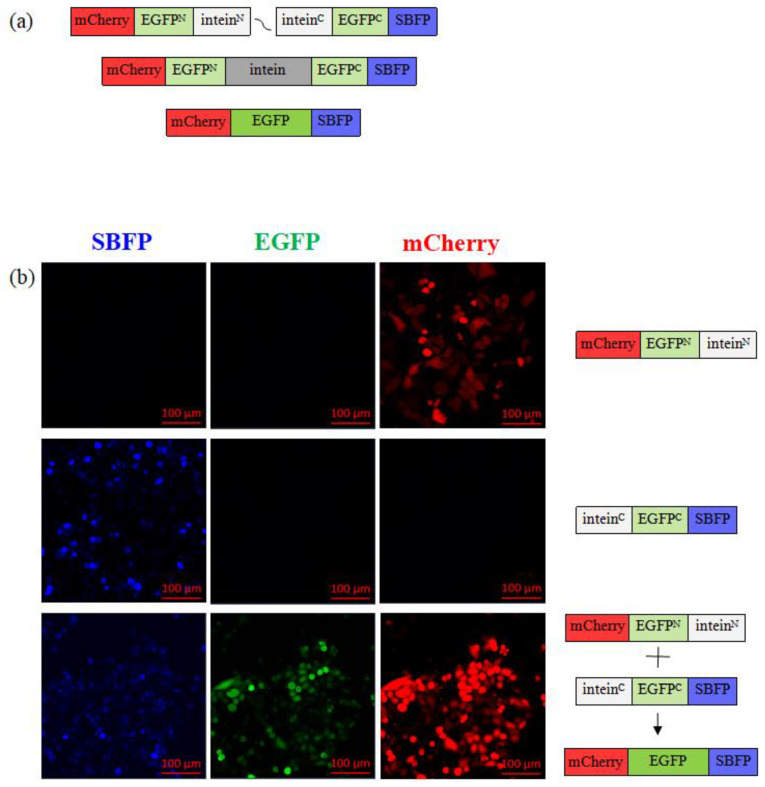
The split EFGP assay involves restoring a fully functional EGFP protein via intein reaction and expression of split EGFP fragments in live cells. (**a**) Precursor fusion proteins are expressed in transiently transfected mammalian cells using a gene-containing plasmid, and each contains a split intein, a non-emitting EGFP fragment, and a reporter protein. Once the HINT structure assembles and intein excision occurs, fragments of EGFP remain ligated, and protein restores its structure, resulting in a detectible green fluorescent signal. (**b**) Representative confocal microscopy images of HeLa cells treated either with p_egfpN_inteinN or p_inteinC_egfpC reporter plasmids, or with both plasmids simultaneously.

**Figure 3 pharmaceutics-16-00617-f003:**
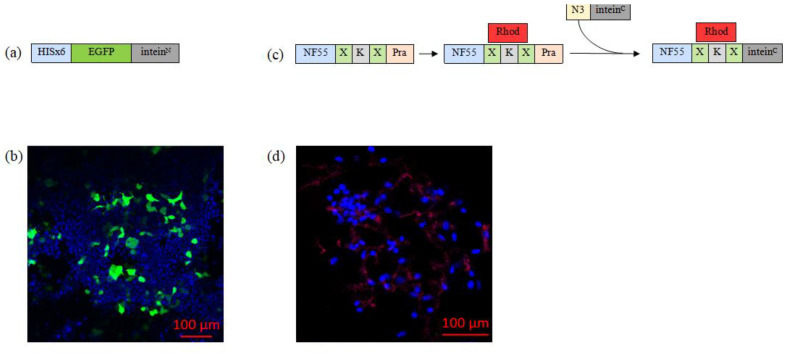
(**a**) Graphical representation of EGFP–Intein^N^ fusion protein. (**b**) Representative confocal microscopy image of HeLa cells that have been transfected with the reporter fusion protein. The cell nuclei were stained with Hoechst (blue). Green is expressed from the delivered plasmid. (**c**) The second graphical representation shows the process of synthesizing the nanoentity chemical moiety, where X—aminohexanoic acid (used as a linker); Pra—propargylglycine; Rhod—carboxytetramethylrhodamine; and N_3_—azide group. (**d**) Representative confocal microscopy images of HeLa cells treated with 5 µM of the chemical compound. Cell nuclei were stained with Hoechst (blue), and intein^C^-activated NF55 was labeled with red fluorescent dye (carboxytetramethylrhodamine).

**Figure 4 pharmaceutics-16-00617-f004:**
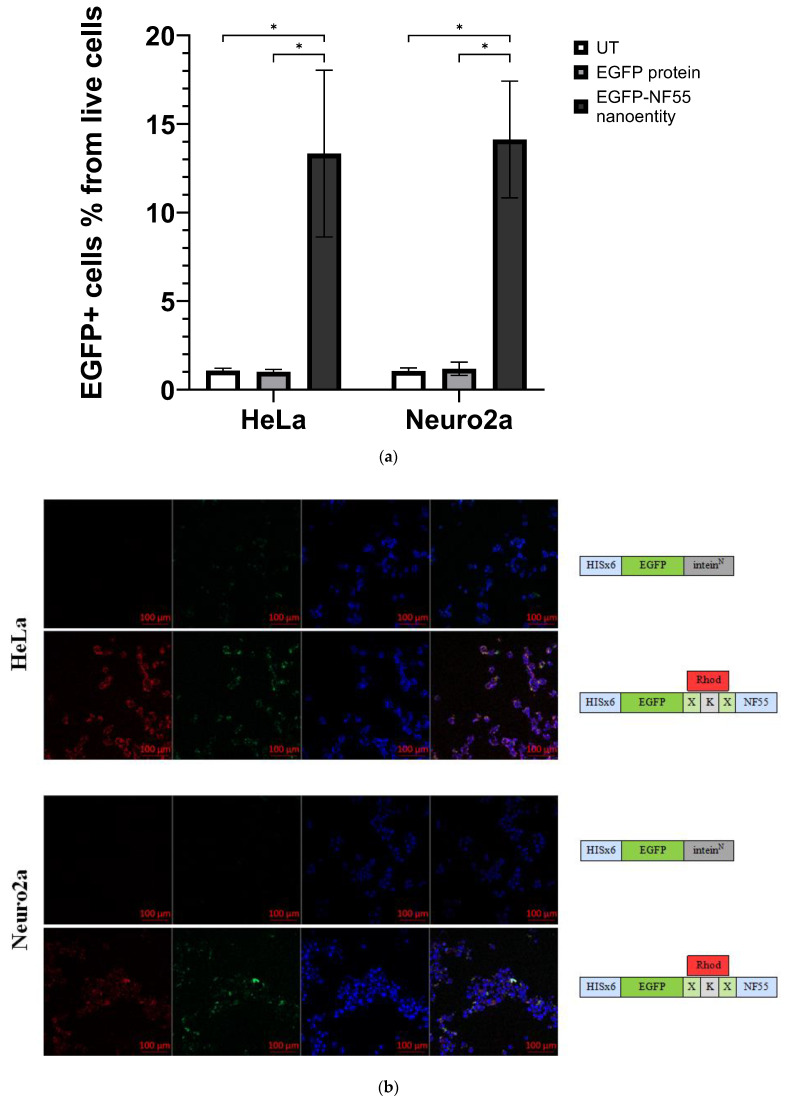
(**a**) The graph shows the percentage of EGFP+ cells from the live cells population assessed via flow cytometry analysis. HeLa or Neuro 2a cells were co-incubated either with EGFP protein or EGFP–NF55 nanoentity (N ≥ 6). Multiple unpaired *t*-test was performed for analysis, where the *p*-value is less than 0.001, summarized with asterisks on the graph. (**b**) Representative confocal microscopy images of cells treated with cell-penetrating bioconjugated nanoentity. Both HeLa and Neuro2a cells were co-incubated with their respective entities. The green signal comes from EGFP, the blue signal comes from cell nuclei stained with Hoechst, and the red signal is from a fluorescent dye called carboxytetramethylrhodamine that is linked to the CPP NF55.

**Table 1 pharmaceutics-16-00617-t001:** Comparison of common bioconjugation methods.

Name	Advantages	Limitations
Click chemistry [32,33]	Highly efficient, specific, and bio-orthogonal	Requires specific reactive groups and is not universally suitable for all biomolecules
Sortase-mediated ligation [32]	Site-specific, efficient, and allows for the incorporation of synthetic peptides into proteins	Limited to those containing a specific motif, potentially requires optimization for various substrates, and has limited activity and stability
Native chemical ligation [34,35,36]	Compatible with a broad range of biomolecules and suitable for large proteins	Necessitates a cysteine residue at the ligation site, and reaction rates may be slow for certain substrates
Protein-trans splicing (NCL analog) [37,38,39,40]	Rapid and can occur in a biological environment without affecting the higher-order structure of the protein	Limited to those containing a specific intein sequence

**Table 2 pharmaceutics-16-00617-t002:** Nanoentity components.

Name	Description	Size
EGFP	Reporter protein component of the nanoentity. Naturally occurring fluorescent protein derived from the jellyfish *Aequorea* *ictoria*.	714 amino acid residues
NickFect 55 (NF55)	Cell-penetrating peptide/vehicle of the nanoentity. Analog of CPP Transportan 10. Stearoyl-AGYLLGO*INLKALAALAKAIL-NH_2_; O*—synthesis continued from the side-chain instead of the alpha-amino group.	21 amino acid residues
Intein^N^	N-terminal split intein fragment. Origin from DNA polymerase III subunit alpha; Anabaena variabilis ATCC 29413.	80 amino acid residues
Intein^C^	C-terminal split intein fragment. Origin from DNA polymerase III subunit alpha; Nostoc sp. ATCC 53789.	39 amino acid residues

**Table 3 pharmaceutics-16-00617-t003:** Split EGFP assay plasmids.

	Name	EGFP Fragment	Spilt Intein Fragment	Control Reporter
First set	p_egfpN_intN	EGFP N-terminal fragment	intein^N^	mCherry (red)
	p_intC_egfpC	EGFP N-terminal fragment	intein^C^	SBFP (blue)
Second set	p_egfp1-10_intN	EGFP 1–10 domains	intein^N^	mCherry (red)
	p_intC_egfp11	EGFP 11 domain	intein^C^	SBFP (blue)

## Data Availability

The raw data supporting the conclusions of this article will be made available by the authors on request.

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
