# Peer review of "Enhancing Cellular Uptake of Native Proteins through Bio-Orthogonal Conjugation with Chemically Synthesized Cell-Penetrating Peptides"

_pharmaceutics, 2024, doi:10.3390/pharmaceutics16050617_

Round 1

Reviewer 1 Report

Comments and Suggestions for Authors

Authors present an interesting study on enhancing protein delivery to cells using an intein-based split protein system. Although the results are promising, some issues need to be addressed before publication.

1. Since the developed tool is intended for further in vivo applications, one of the main concerns is the solubility of NF55 with a stearoyl group. It is evident that this substance may not be soluble in water buffer systems at sufficient concentrations. Authors should discuss this issue in detail and propose possible solutions, including alternative cell-penetrating peptides.

2. Flow cytometry data should be supported with dot plots and histograms showing fluorescence intensity. Authors should provide flow cytometry data with a gating strategy, dot plots of FSC-SSC, and flow cytometry histograms in the fluorescent channel corresponding to EGFP fluorescence. They should also describe in detail how the percentage of cells was calculated.

Author Response

Dear Reviewer, Thank you for your thoughtful and constructive comments on our manuscript. We sincerely appreciate the time and effort you have dedicated to providing feedback. Please find our response for your comments down below.
  1. Since the developed tool is intended for further in vivo applications, one of the main concerns is the solubility of NF55 with a stearoyl group. It is evident that this substance may not be soluble in water buffer systems at sufficient concentrations. Authors should discuss this issue in detail and propose possible solutions, including alternative cell-penetrating peptides.

Answer:

We appreciate your attention to the solubility concerns regarding NF55, particularly in the context of future in vivo applications. It is indeed crucial to ensure the solubility of the developed tool for effective translation into in vivo settings. Research of our group is focused on utilization of cell-penetrating peptides for in vitro and in vivo applications. Based on our expertise we selected NF55 as a cell-penetrating peptide as the most potential for in vivo applications. We have evidence that NF55 has low toxicity and is highly efficient in in vivo settings.  (http://dx.doi.org/10.1016/j.jconrel.2016.09.022,  https://doi.org/10.1038/s41598-019-56455-2, https://doi.org/10.3390/pharmaceutics15030883).

While NF55, with its stearoyl tail, is originally designed to be soluble and validated as such, we acknowledge that concerns may arise regarding its solubility at higher concentrations, although these concentrations are beyond intended biological use. However, it's important to note that in our study, we utilized NF55 within concentrations where solubility was not an issue. We have carefully optimized the experimental conditions to ensure the effective solubility and functionality of NF55 in our experimental setup.

  1. Flow cytometry data should be supported with dot plots and histograms showing fluorescence intensity. Authors should provide flow cytometry data with a gating strategy, dot plots of FSC-SSC, and flow cytometry histograms in the fluorescent channel corresponding to EGFP fluorescence. They should also describe in detail how the percentage of cells was calculated.

Answer:

Thank you for your attention to the presentation of the flow cytometry data and providing requests for additional supporting information, including dot plots, histograms, and a description of the gating strategy. We would like to clarify that the flow cytometry data is indeed included in sections 2.6 and 2.7. It's worth noting that these conditions were applied consistently to each conducted flow cytometry experiment. We repeated flow cytometry measurements in three separate experiments, with three replicates for each condition. To avoid overloading the reader with data, we did not include histograms or dot plots from every experiment in the original manuscript version. However, in response to the reviewer's suggestion, we have added Figures S4 and S5 as representative flow cytometry plots demonstrating the gating strategy.

Thank you once again for your insightful comments and constructive feedback. We have carefully considered your suggestions and believe that our revisions have improved the quality of our manuscript. We look forward to your assessment of the revised version and hope for the opportunity to contribute to the advancement of research in the field.

Reviewer 2 Report

Comments and Suggestions for Authors

This manuscript "Enhancing Cellular Uptake of Native Proteins Through Bioorthogonal Conjugation with Chemically Synthesized Cell Penetrating Peptides” describes the potential of protein transduction for treating genetic and metabolic disorders, emphasizing the importance of precise delivery for therapeutic efficacy. It introduces a novel approach utilizing a bioorthogonal conjugation method and protein trans-splicing to enhance the cell-penetrating abilities of cargo proteins, aiming to overcome challenges such as endosomal entrapment and degradation for effective therapeutic delivery.

Few comments are as follows:

Comment 1: The introduction spends a significant amount of time on background information about protein transduction and CPPs. It could benefit from a stronger focus on the specific research question and the novelty of the proposed method (protein trans-splicing for CPP conjugation).

Comment 2: Sentence 117 provides information about obtaining split-intein sequences from a previous study, but it lacks clarity regarding the specific plasmids used for cloning. Clarifying the origin or catalog numbers of these plasmids would enhance reproducibility.

Comment 3: While the synthesis of NF55 peptide is described in detail in sentences 123-136, the description could be improved by providing more specific reaction conditions, such as reaction times and temperatures.

Comment 4: Line 152 mentions the purification of the NF55-XK(Rhod)X-inteinC conjugate, but it does not specify how the conjugation was confirmed. Including details on the characterization techniques used to validate the conjugation, such as mass spectrometry or spectroscopy, would strengthen the methodology.

Comment 5: Line 180 describes the seeding of cells for confocal microscopy, but it lacks information on the specific staining or labeling methods used to visualize the nanoentities or fusion proteins.

Comment 6: The validation of nanoentity assembly is briefly mentioned in sentences 332-337, but more details on the experimental setup and criteria for successful conjugation would strengthen this section. Including information on how changes in peaks detected with UPLC confirm successful conjugation would provide experimental evidence to support the claims.

Comment 7: While the cellular uptake of the nanoentity is assessed through confocal microscopy and flow cytometry analysis in sentences 340-343, it would be beneficial to include quantitative data on the efficiency of transduction, such as the percentage of cells showing EGFP signal or the mean fluorescence intensity.

Comments on the Quality of English Language

Minor editing required

Author Response

Dear Reviewer, Thank you for your thoughtful and constructive comments on our manuscript. We sincerely appreciate the time and effort you have dedicated to providing feedback. Please find our response for your comments down below.
  1. Comment 1: The introduction spends a significant amount of time on background information about protein transduction and CPPs. It could benefit from a stronger focus on the specific research question and the novelty of the proposed method (protein trans-splicing for CPP conjugation).

    Answer:

    We acknowledge your insightful suggestion regarding the introduction section of our manuscript. We understand the importance of maintaining a balance between providing background information and emphasizing the novelty of our proposed method. While it is crucial to articulate the specific research question and the novelty of the protein trans-splicing approach for CPP conjugation, it is equally essential to provide a comprehensive understanding of the underlying concepts of protein transduction and CPPs to contextualize our work effectively.

    While a significant part of introduction provides necessary background information, lines 80-94 are specifically focused on explanation of the protein trans-splicing technique itself, and novelty of the protein-trans splicing method.

    Comment 2: Sentence 117 provides information about obtaining split-intein sequences from a previous study, but it lacks clarity regarding the specific plasmids used for cloning. Clarifying the origin or catalog numbers of these plasmids would enhance reproducibility.

    Answer:

    We obtained the sequences of inserts from the study by David et al. (2015), as mentioned in lines 117 and 118 of the original manuscript. Specifically, lines 118 and 119 provide details about plasmids containing these inserts ordered from the service provider. For clarity and reproducibility, we have listed the plasmids and insert sequences used in our study in Supplementary Figure S1.

    Comment 3: While the synthesis of NF55 peptide is described in detail in sentences 123-136, the description could be improved by providing more specific reaction conditions, such as reaction times and temperatures.

    Answer:

    We appreciate the reviewer's attention to the synthesis protocol for NF55. It is crucial to acknowledge that while reaction times and temperatures indeed play a role, they are predominantly governed by the specifications of the peptide synthesizer, which can vary even within identical setups. Moreover, based on our extensive experience and empirical observations, we are confident that the synthesis parameters do not significantly alter the properties of NF55. We want to assure the reviewer that the details provided in the Methods section are comprehensive and adequate to replicate the synthesis of NF55 by any other party. It is important to note that the synthesis of NF55 follows a standardized machine protocol with minimal deviations. We trust that these clarifications alleviate any concerns regarding the reproducibility of our synthesis methodology, and we remain committed to ensuring the transparency and accessibility of our experimental procedures.

    Comment 4: Line 152 mentions the purification of the NF55-XK(Rhod)X-inteinC conjugate, but it does not specify how the conjugation was confirmed. Including details on the characterization techniques used to validate the conjugation, such as mass spectrometry or spectroscopy, would strengthen the methodology.

    Answer:

    As indicated in lines 150-151 and 312-313 the occurrence of NF55-XK(Rhod)X-inteinC conjugate was confirmed by UPLC. Figure S3 depicts a reduction in the amount of inteinC, which was initially in excess compared to NF55-XK(Rhod)XPra, and the disappearance of NF55-XK(Rhod)XPra from the reaction mixture. Furthermore, the emergence of a new compound (NF55-XK(Rhod)X-inteinC conjugate) can be observed. In response to the reviewer’s suggestion we have modified Figure S3 description including details and explanation for clarity.

    Comment 5: Line 180 describes the seeding of cells for confocal microscopy, but it lacks information on the specific staining or labeling methods used to visualize the nanoentities or fusion proteins.

    Answer:

    As described in the main text, the plasmids utilized in our study expressed reporter fluorescent proteins. Additionally, the NF55-InteinC component was labeled with a dye called carboxytetramethylrhodamine (Rhod). This labeling approach obviated the need for additional staining or labeling of cells prior to visualization. In instances where it did not compromise experimental outcomes, the nucleus was stained with Hoechst, as indicated in the figure descriptions.

    Comment 6: The validation of nanoentity assembly is briefly mentioned in sentences 332-337, but more details on the experimental setup and criteria for successful conjugation would strengthen this section. Including information on how changes in peaks detected with UPLC confirm successful conjugation would provide experimental evidence to support the claims.

    Answer:

    In response to the reviewer’s suggestion, we have added Figure S6 with experimental evidence to support the claim of successful conjugation.

    Comment 7: While the cellular uptake of the nanoentity is assessed through confocal microscopy and flow cytometry analysis in sentences 340-343, it would be beneficial to include quantitative data on the efficiency of transduction, such as the percentage of cells showing EGFP signal or the mean fluorescence intensity.

    Answer:

    We appreciate the reviewer's suggestion to include quantitative data on the efficiency of transduction. For flow cytometry analysis the percentage of EGFP positive cells is provided in Figure 4a. In response to the reviewers’, we have added Figure S7, where we provided data on the percentage of EGFP+ area, which serves as a measure of transduction efficiency. This data complements the confocal microscopy provided in the main text.

Thank you once again for your insightful comments and constructive feedback. We have carefully considered your suggestions and believe that our revisions have improved the quality of our manuscript. We look forward to your assessment of the revised version and hope for the opportunity to contribute to the advancement of research in the field.

Round 2

Reviewer 1 Report

Comments and Suggestions for Authors

The authors responded to all requests, I recommend accepting the publication in its current form.